# Research sites get closer to field camps over time: Informing environmental management through a geospatial analysis of science in the McMurdo Dry Valleys, Antarctica

Stephen M. Chignell[1]*, Madeline E. Myers[2¤], Adrian Howkins[3], Andrew G. Fountain[4,5]

**1** Institute for Resources, Environment and Sustainability, University of British Columbia, Vancouver, British Columbia, Canada, **2** Department of Geology and Geophysics, Louisiana State University, Baton Rouge, Louisiana, United States of America, **3** Department of History, University of Bristol, Bristol, United Kingdom, **4** Department of Geology, Portland State University, Portland, Oregon, United States of America, **5** Department of Geography, Portland State University, Portland, Oregon, United States of America

¤ Current address: Department of Geography and Planning, Queen's University, Kingston, Ontario, Canada
* steve.chignell@ubc.ca

**Data Availability Statement:** All relevant data are within the paper and its Supporting information files.

## Abstract

As in many parts of the world, the management of environmental science research in Antarctica relies on cost-benefit analysis of negative environmental impact versus positive scientific gain. Several studies have examined the environmental impact of Antarctic field camps, but very little work looks at how the placement of these camps influences scientific research. In this study, we integrate bibliometrics, geospatial analysis, and historical research to understand the relationship between field camp placement and scientific production in the McMurdo Dry Valleys of East Antarctica. Our analysis of the scientific corpus from 1907–2016 shows that, on average, research sites have become less dispersed and closer to field camps over time. Scientific output does not necessarily correspond to the number of field camps, and constructing a field camp does not always lead to a subsequent increase in research in the local area. Our results underscore the need to consider the complex historical and spatial relationships between field camps and research sites in environmental management decision-making in Antarctica and other protected areas.

## Introduction

By many measures, the Antarctic continent is among the most protected environments anywhere on the planet [1]. The 1991 Protocol on Environmental Protection to the Antarctic Treaty (commonly known as the Madrid Protocol) prohibited economic mineral activities, introduced extensive measures to prevent the introduction of non-native species, and established a Committee for Environmental Protection to assess the environmental impact of human activities in Antarctica. An important tool of the committee is the environmental impact assessment, which has been used in Antarctica since the mid-1970s [2–5]. In common with their use in other parts of the world, environmental impact assessments offer a template

**Funding:** This research has been supported by the National Science Foundation, Office of Polar Programs (award numbers 1443475 [AH and AGF] and 1637708 [AH], website: https://nsf.gov/div/index.jsp?div=OPP) and by the British Academy (award number KF3/100152 [AH and AGF], website: https://www.thebritishacademy.ac.uk/). Geospatial data support for this work was provided by the Polar Geospatial Center under NSF-OPP awards 1043681 and 1559691. The funders had no role in study design, data collection and analysis, decision to publish, or preparation of the manuscript.

**Competing interests:** The authors have declared that no competing interests exist.

for conducting a cost-benefit analysis of the negative environmental impact of an activity versus the positive scientific and logistical gains.

The construction of semi-permanent field camps (Fig 1) for supporting science in ice-free regions offers a particularly prominent example of the cost-benefit analyses involved in regulating human activities in Antarctica. By semi-permanent, we mean camps with structures used over multiple years. In contrast, temporary camps, used for days to months, are removed prior to winter. In general, field camps in ice-free regions of Antarctica are understood to have negative impacts on the local environment, including local soil compaction [6], bacterial contamination [7], fossil fuel spills [8–11] and burning [12, 13], introduction of invasive species [14], and the diminution of wilderness values [15, 16]. The period since the signing of the Madrid Protocol has seen the removal of many field camps from Antarctica as part of efforts to reduce human impacts on the Antarctic environment, although there are concerns that the overall human footprint is increasing [17, 18].

Environmental managers conducting cost-benefit analyses of field camps require both an understanding of a camp's scientific value as well as its environmental impact, including the associated activities in the surrounding area. However, in contrast to environmental impact, the value of field camps to stimulate scientific research has received relatively little attention in Antarctica and elsewhere. Pertierra et al. (2017) consider scientific productivity in their study of field camps in the Byers Peninsula, but their primary focus was environmental impact [17]. In addition to journal rankings and citation metrics, scientific productivity is often measured by the number of research publications [20–22]. While there are many other factors influencing the productivity of specific camps (e.g., funding, personnel, remoteness), publication frequency provides a consistent metric that can be computed and compared across the full history of scientific activity. Even still, assigning specific locations to the research described in

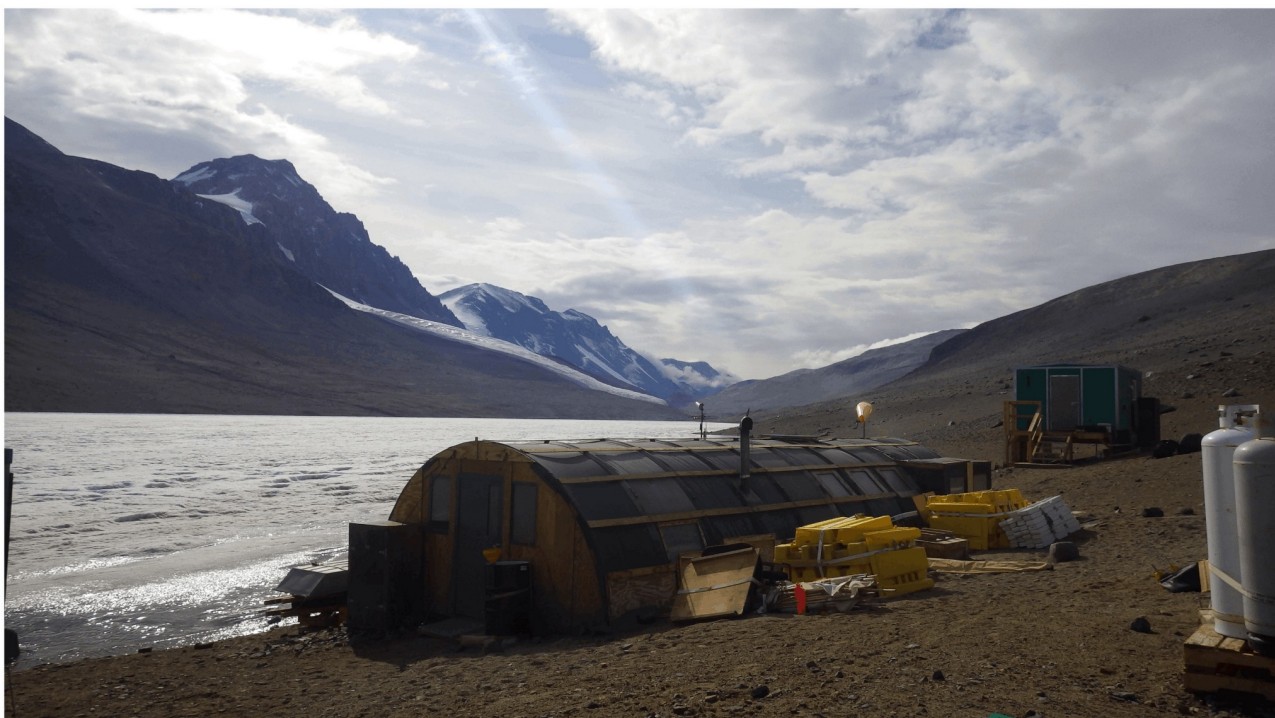

**Fig 1. Lake Bonney Camp (New), a semi-permanent field camp in the McMurdo Dry Valleys, Antarctica.** Photo credit: Stephen Chignell, 2016, available in the McMurdo Dry Valleys History Archive (record #MCMEH-P000099) [19].

each publication can be challenging and time-consuming. Such work requires different forms of expertise including knowledge of local geography and logistics, as well as a qualitative understanding of field practices in various scientific disciplines.

The McMurdo Dry Valleys (MDV), located along the western shore of the Ross Sea in East Antarctica, offer an ideal case study for examining the association between field camp placement and scientific activity. The region has a total area of about 22,700 km$^2$, and an ice-free area of about 4,500 km$^2$ [23], making it the largest ice-free area on the continent [24, 25]. The MDV are of great scientific interest due to their exposed geology and matrix of soils, lakes, glacial streams [26, 27], which serve as terrestrial analogs for Mars [28–30] and possible sentinels of global climatic changes [31–33]. The MDV are surrounded by the Antarctic ice sheet on three sides and the Ross Sea on the fourth, which define the area geographically and bound the area of scientific activity. The region's geographical clarity is further highlighted by its designation as an Antarctic Specially Managed Area (ASMA) by the Antarctic Treaty [34]. Several of the environmental management protocols currently in force throughout Antarctica were developed in the MDV [4, 5, 35]. Since the late 1950s, the MDV have been an important center of Antarctic science, hosting research every summer season and over the course of three winters. During this period, 16 semi-permanent field camps have been constructed—primarily for conducting scientific research [36]. Seven of these camps were removed between 1958 and 2016.

The ebb and flow of established field camps and scientific activity offers an opportunity to examine how research activity in the MDV responds to camp establishment and removal. Given the challenging environmental conditions and the desire of researchers and national research programs to make efficient use of time and resources, it seems likely that the construction of a new field camp would encourage researchers to study areas nearby the camp. This might be spurred further by the need to reduce environmental impact following the adoption of the Madrid Protocol in 1991. It also seems likely that the increased logistical capacity provided by a field camp would support more complex, instrument-intensive studies and become a draw for new investigators and new research projects. In this way, field camps might act as centers of gravity that 'pull' researchers (and their study sites) toward them over time. Following this line of reasoning, one would also expect this attractive force to disappear after the removal of the field camp and researchers' attention to shift toward other regions.

Using bibliometrics, geospatial analysis, and historical research, we examine the relationship between the timing and location of semi-permanent field camps to the patterns of scientific research in the MDV. We test two hypotheses. First, study sites become progressively closer to field camps over time. Second, the number of publications, a proxy of scientific activity, increases after a field camp is established and decreases following its removal. We then consider the limitations and implications of our results for environmental management in the MDV, and discuss the potential for adapting our approach to other regions.

## Materials and methods

### Data acquisition

**Bibliometric data.**   To identify field sites of scientific activity, we compiled and examined publications related to the MDV up through 2016. While our approach does not qualify as a systematic review or meta-analysis according to the Cochrane requirements [37], we conducted an extensive review, combining in-depth descriptive and inferential statistical techniques with historical context from the literature. We outline our approach below and also include a flow diagram [38] detailing our review and screening process (S2 File). Between 1978 and 1995, the New Zealand (NZ) Antarctic Program published three volumes of a

Bibliography of International Dry Valley Publications [39–41]. These include all mainstream academic literature (e.g., journal articles, books, conference proceedings) as well as 'grey' literature (e.g., theses/dissertations, institute reports, unindexed journal articles) published between 1959 and 1994. We acquired paper copies of these volumes and digitized all references (n = 1,569). To supplement and update that bibliography, we queried the Web of Science Core Collection (accessed July 25, 2017) for MDV-related publications using the 'Topic' field tag, which searches title, abstract, and keywords. We constrained the search to the years 1900–2016 and used the following query: (TS = ("McMurdo Dry Valleys") OR TS = ("Taylor Dry Valley") OR TS = ("Wright Dry Valley") OR TS = ("Victoria Dry Valley") OR TS = ("Taylor Valley") OR TS = ("Wright Valley") OR TS = ("Victoria Valley") OR TS = (Dry Valley*) OR TS = (Ice-free Valley*)) AND TS = (Antarctica). After downloading the resulting references (n = 1,257), we conducted an initial screening of all references from the published bibliography and Web of Science and removed obvious duplicates (n = 84). We then conducted a more in-depth screening to identify and remove additional duplicates, such as those with slight variations in titles (n = 22).

We read the texts of each of the remaining references (n = 2,720) to assess their eligibility for identifying locations where fieldwork occurred in the MDV. We excluded references for which we could not find a full-text through internet searches or university libraries, as well as those written in non-English languages. We also excluded review or synthesis articles based on secondary data sources, as well as remote sensing and aerial surveys that did not have a field sampling component. We found the methods, acknowledgements, and captions most useful in this process. Of the full-texts assessed, 1,486 (55%) satisfied our criteria, with 628 having one study site, and 658 more than one study site. We excluded the remaining 200 publications because they did not contain enough information to locate the study sites. We then extracted and compiled the names of the study sites for publications containing study site information (e.g., 'Taylor Glacier', 'Lake Vanda', 'Commonwealth Stream', and so on).

**Geospatial data.** We acquired geographic information system (GIS) vector layers of internationally recognized Antarctic features (e.g., glaciers, lakes, streams) from the MDV Long-Term Ecological Research Project [42]. We augmented this with point data from the U. S. Geographic Names Information System (https://www.usgs.gov/core-science-systems/ngp/board-on-geographic-names/antarctic-names) which we customized based on resources at the Polar Geospatial Center at the University of Minnesota and our own knowledge of the area. We then matched study sites identified in the publications to a point location based on the GIS layers (n = 284). For linear features such as streams, we used the location of the stream gauge. For ungauged streams or those with multiple gauges, we used the midpoint of the stream feature. For polygons and multipoint features such as lakes and glaciers, we used the centroid of the polygon. These points are approximations, since the actual locations of study sites would vary for each feature and study. For example, field sampling for a study on glacial discharge would tend to take place near the edge of the glacier, whereas a study sampling cryoconite holes would be more centrally located. Based on the MDV Long-Term Ecological Research Project GIS data, the average area of the 130 named glaciers is 46.7 km$^2$, the average area of the 28 named lakes is 1.2 km$^2$, and the average length of the 28 named streams is 3.1 km. The latter calculation excludes the Onyx River, which has a length of 41.6 km. Given that most field sampling of the streams takes place at gauges, and the relatively small size of the lakes, the four or five largest glaciers in the MDV represent the greatest source of uncertainty in our analysis. Although these glaciers lower our precision in some areas, it is reasonable to assume the feature centroids are accurate markers of study locations at the regional scale. Our approach thus provides an imperfect but consistent way of comparing study site locations with uncertain and variable location information across multiple types of

features. Using historical photographs, field surveys, and scientific reports [19, 36], we compiled information on the locations of semi-permanent field camps (henceforth, 'field camps') and their dates of operation. We converted all GIS data into the South Pole Azimuthal Equidistant projection (WGS 1984 datum).

Research activity in the MDV overwhelmingly takes place in the austral summer months (November–February) because the camps are closed in winter (except for a few special winter campaigns). Therefore, we assigned establishment and removal dates of field camps to the year the summer season ends (e.g., the old Vanda Station was built over the 1968–69 season and removed during the 1994–95 season, so we set its start and end dates to 1969 and 1995, respectively). We assumed that research activity within a 5 km radius of a field camp used that camp. We chose this distance based on our own fieldwork experiences in the area and common travel times to and from field study sites. For comprehensive inclusion of all research activity, we assumed that research within a 20 km radius around each camp may have an association with that camp, acknowledging that overlap between camp radii may occur. Based on the historical record, 20 km is the approximate upper limit that someone would travel to a research site (e.g., trips from the old Vanda Station to Don Juan Pond and The Labyrinth) [43]. Although subjective, these distances have the benefit of enabling comparisons between 'local' (< 5 km) and 'distant' (5–20 km) research at each field camp (Fig 2). Henceforth, we refer to the total area within 20 km of a field camp as the 'surrounding area'.

## Data analysis

To provide an overview of how research output corresponds to field camp availability over time, we first compared temporal trends in publication frequency versus the number of field camps present in the MDV. We then examined the same data geospatially, computing broad-scale metrics to characterize spatial clustering in research activity and its overlap with the areal extent of field camps. Finally, we conducted a detailed proximity analysis, measuring changes in the distance between field camps and study sites at the scale of the whole MDV and for individual camps.

Using the bibliometric data and field camp construction/removal dates, we counted the total number of publications and total number of camps for each year. To identify statistically significant ($p < 0.05$) trends and their changes over time, we used MATLAB R2019b software to conduct a changepoint analysis on both data sets. This identifies points within a time series when the statistical properties change, and then segments the time series based on those changepoints. We followed the approach of Edelhoff et al. (2016), who identified changepoints in animal movement by iteratively dividing the time series into segments and calculating the sum squared error of the best fit line of each segment [45]. We chose the changepoints which produced the minimal sum squared error on either side for deriving Sen's slope between changepoints and conducting Mann-Kendall tests for significance. Sen's slope is the median of slopes through all pairs of points and is fairly resistant to outliers [46].

To measure how publication frequency and field camp availability varied in time and cartesian space, we computed centrographic statistics for the cumulative dataset and for each year using ArcMap (version 10.7) GIS software. This included the geometric mean center and standard distance of the study site locations, both weighted by the number of publications at each site. The mean center is the average x- and y-coordinate of all features in a dataset. The standard distance is a measure of how concentrated or dispersed features are around the mean center (similar to how a standard deviation measures the distribution of values around the statistical mean). Standard distance is visualized as a circle, with the radius equal to the standard distance (at one standard deviation, in our case) [47].

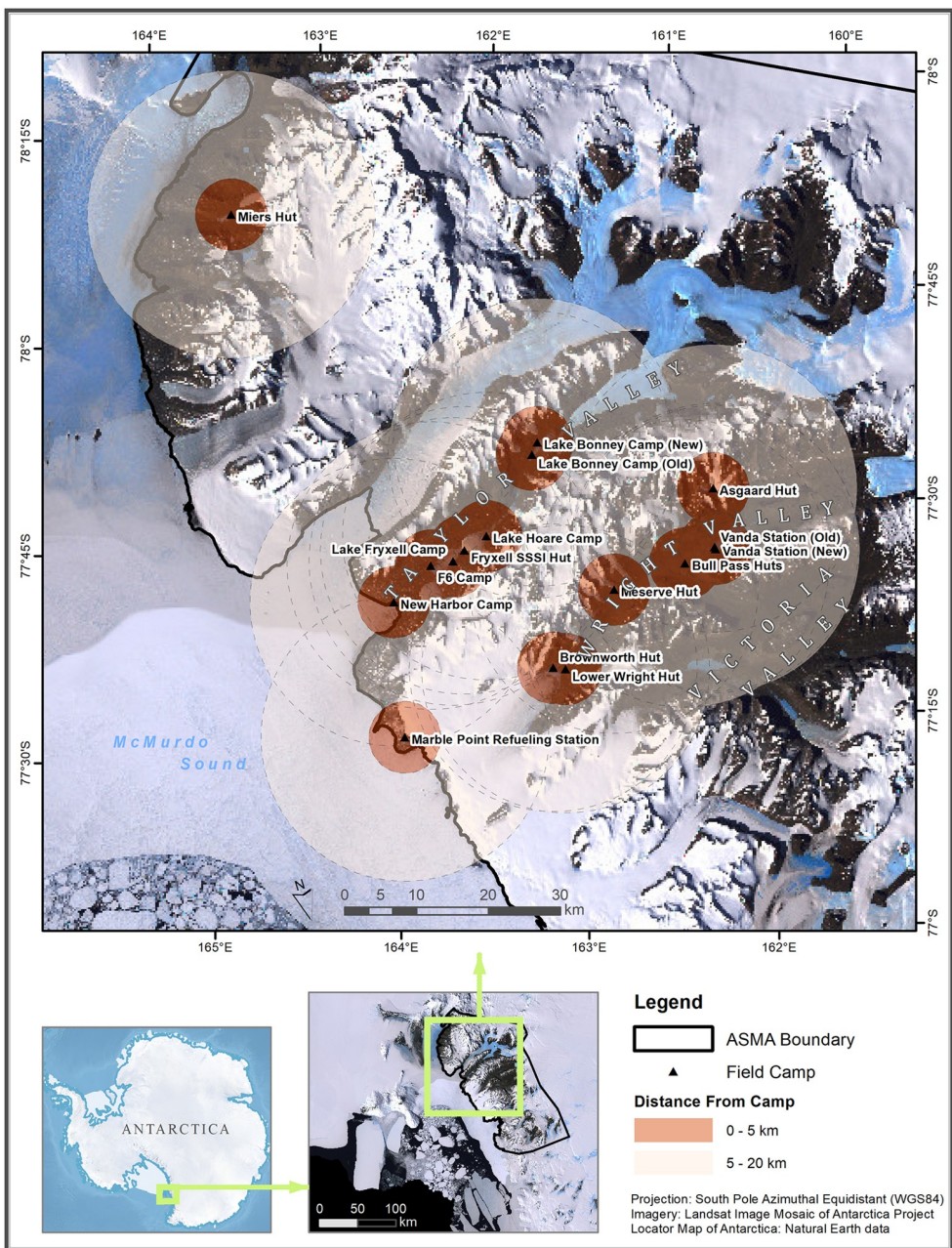

**Fig 2. Map of the McMurdo Dry Valleys and associated Antarctic Specialty Managed Area (ASMA).** Locations of the 16 historic and current semi-permanent field camps are shown with 5 km ('local') and 20 km ('distant') circular buffers, overlain on satellite imagery [44].

We defined the areal extent of field research across the MDV by a minimum bounding polygon (convex hull) around the study sites, extended by an additional 20 km buffer on all sides (corresponding to our definition of 'surrounding area'). We created such a bounding polygon for the cumulative study sites and for each year. Using the same procedure, we also created bounding polygons based on the field camp locations, cumulatively and for each year.

To examine the overlap between areas studied and areas serviced by field camps, we calculated what percent of the study site standard distance circle fell outside the field camp

bounding polygon (i.e., the proportion of field research that took place beyond the area serviced by field camps). This provides a measure of overlap that is robust against spatial outliers in the study site data (e.g., a single remote study site 'pulling' the bounding polygon over a much larger area). We computed the percent overlap for each year, and also conducted changepoint analyses and Mann-Kendall tests on the time series of these data. To prevent distortion effects, we made all area calculations geodesically.

Centrographic statistics are often used in conjunction with hotspot analyses, which help to visualize how the intensity of a phenomenon varies across an area [48]. We used ArcMap's Kernel Density tool to map hotspots of research activity based on publication counts at each study site location. Kernel density estimation (KDE) calculates the density of features (in this case weighted by the number of publications at each study site) in a spatial neighborhood around those features, producing a raster surface of density values across the landscape. ArcMap uses a quartic kernel formula, and we set the search radius (or 'bandwidth') to the value determined by the optimal bandwidth function [49] (Protocol A in S1 File). To generate a cumulative map of publication density, we created a KDE surface using the full study site data for all years. We also created a KDE surface for each year in order to visualize how publication density in the MDV has changed over time.

To determine whether, at the scale of the MDV, the distance between study sites and field camps has decreased over time, we computed the Euclidian distance between each study site and the nearest camp during the year of publication. We conducted a Mann-Kendall test on the Euclidian distance and standard distance time series to check for a monotonic trend. Where trends were statistically significant, we computed Sen's slope ($p < 0.05$). To check for random associations, we generated an additional dataset of 284 randomly distributed sites within the MDV ASMA boundary, which matches the number of unique study sites we identified in the literature. We then calculated annual mean distance from each random site to the nearest field camp so that any observed trends in distance were the result of adding and removing camps, rather than variability in study site proximity.

Following this, we detrended the annual average distance to the nearest field camp to the randomized annual average distance to the nearest field camp using the simple differencing method. We then normalized the data on a 0–1 scale to create a distance metric which we refer to as the 'normalized distance'. This metric indicates how much further a researcher traveled to get to a study site relative to the average distance required based on field camp availability. Normalized distance values closer to 1 indicate a greater distance traveled to get to a study site. We then conducted a linear changepoint analysis on the distance metric to identify statistically significant trends.

To analyze historical changes in study site distance within and among individual field camps, we grouped the study site data into classes. First, we identified all publications that included study sites within the surrounding area of each camp (20 km radius). Within this circle, we created circular buffers at < 5 km and 5–20 km from each camp. Next, we divided the duration (in years) of each camp into quartiles. When defining quartile lengths, we followed a consistent method to avoid splitting field seasons which are summer occurrences spanning the change in calendar year (Table A in S1 Dataset). For each camp, we counted the number of publications per quartile located within each of the previously defined buffer distances. If a publication (i.e., study site) was located within the buffers of multiple camps, we counted it for each relevant camp.

To compare periods when a camp was in place with the periods when it was not, we repeated the same counting procedure for four additional quartiles of the same duration as the initial quartiles. For camps that remained in place through 2016, we counted publications for

four quartiles prior to the construction of the camp. For camps that have been removed, we counted publications for two quartiles before and two quartiles after the camp existed.

Although our study covered the full temporal range of MDV science, our data screening process resulted in just three publications from the years prior to 1958 with enough location information. These comprised the early 20[th] century studies of Griffith Taylor and others of the 'Heroic Era' in Antarctic science [50]. Since this is an important part of the history of the MDV, we included these years in analyzing publication frequency near a camp location over time. However, given the very low number of early publications, we began the analysis based on distance to an existing camp in 1958, the year after the first camp was constructed.

## Results

The combination of linear changepoint analyses and Mann-Kendall tests revealed three statistically significant periods for publication frequency and one for the number of field camps (Fig 3) (See S1 Dataset for detailed results of all statistical tests). From 1958 to 1979, publication frequency increased by approximately four per year. This was followed by a decreasing trend of two per year until 1997, at which point the trend then shifted upward again, with publication frequency increasing by approximately 3.5 per year until 2016.

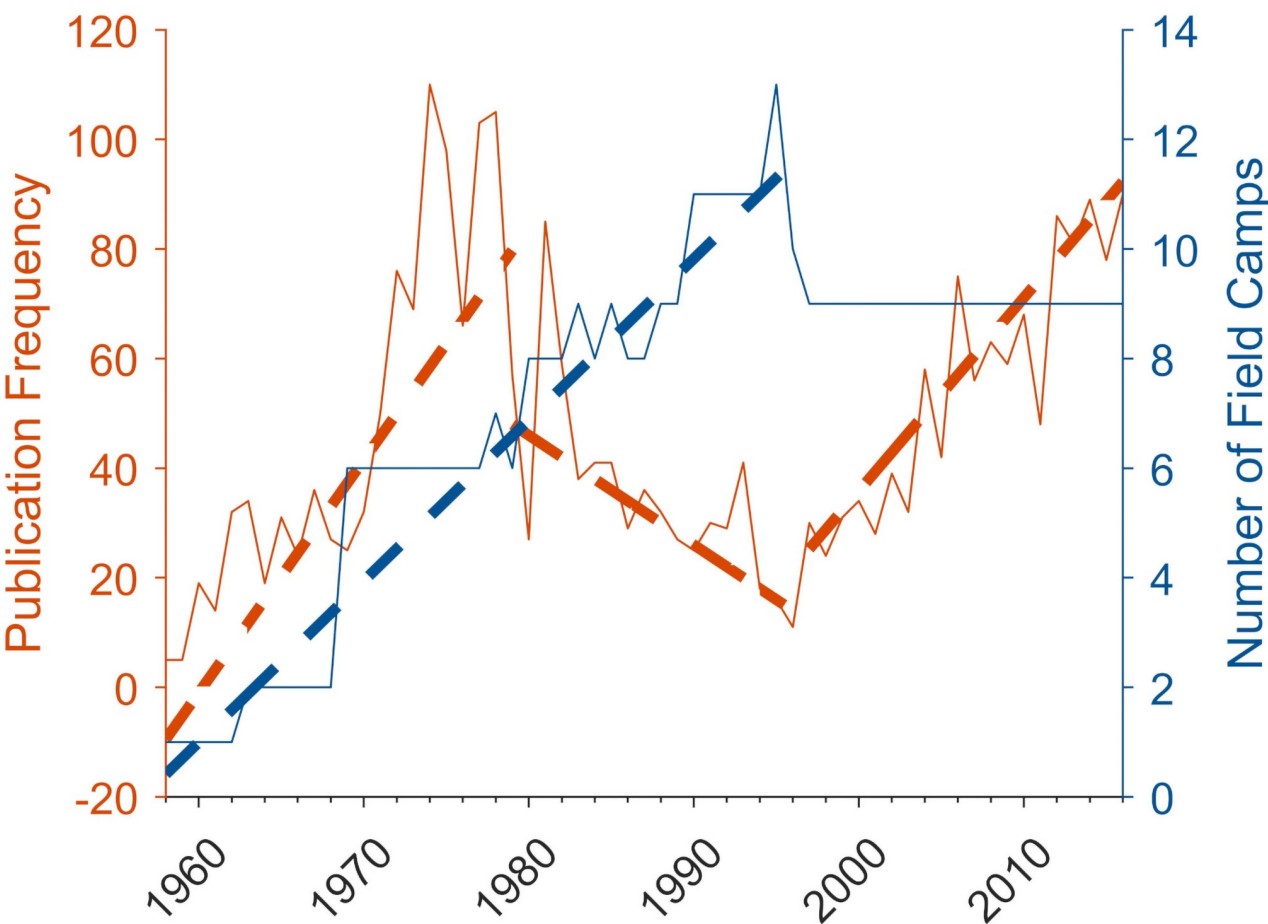

**Fig 3. Number of publications and field camps by year since the construction of the first field camp in 1958.** Dashed lines indicate statistically significant Sen's slopes between changepoints. For publication frequency, p-values from left to right are < 0.001, < 0.01, and < 0.001. For number of field camps, p < 0.001.

The only changepoint for the number of field camps occurred in 1996. The increasing trend from 1958 to 1996 shows the addition of a new camp approximately every three years. The most notable increase in field camps occurred in 1968 with the addition of four camps in Wright Valley. Several camps were removed in 1996, and from then on, the number of field camps remained constant.

These results suggest three periods of coincident changes in MDV scientific research and field camp availability: 1) simultaneous increase in publications and camps (1958–1979), 2) decline in publications during continued camp construction (1979–1996), and 3) increase in publication frequency with no change in camp availability (1996–2016).

The annual centrographic statistics and hotspot maps indicate that the extent and intensity of field research has varied considerably over the historical record (S1 Video). Sporadic research activity has occurred throughout the ASMA boundaries, but cumulative publication density is greatest in Taylor and upper Wright Valleys (Fig 4). The cumulative KDE and standard distance circle also indicate that most research has taken place within the minimum bounding polygon of all 16 field camps. The most intense hotspots occurred in the areas of Lakes Fryxell, Hoare, Bonney (in Taylor Valley), and Vanda (in Wright Valley).

The overlay analysis of the standard distance and field camp bounding polygon areas (see purple circle, white dashed line, respectively in Fig 4) revealed a striking temporal pattern. The annual analysis of these data (Fig 5a) shows that in 1958, 97% of the study site standard distance circle fell outside the area of the field camp bounding polygon. In other words, most field research took place well beyond the area serviced by field camps. This percentage decreased rapidly over the next 20 years, at the same time as the number of field camps and publications were both increasing (i.e., more field camps, more research activity, and greater overlap in their areas). This trend continued to 1977, when 0% of the standard distance circle fell outside the field camp bounding polygon. Our analysis identified a statistically significant changepoint in 1977 and monotonic trend leading up to 1977. No significant trend appeared after 1977, but brief spikes occurred in the mid-1980s, late 1990s, and early 2010s (never reaching above 41%).

Across the entire MDV, from 1958 to 2016, mean annual distance between study sites and the nearest field camp decreased by 15 km (24%), while the standard distance decreased by 10 km (22%). During the same period, the annual average distance to the nearest camp for a set of randomly distributed camps decreased by 26 km (35%, Fig 5b), which implies the increasing proximity of study sites to field camps is a consequence of field camps placement based on logistic, rather than scientific reasons. On average, study sites have become closer to field camps and less dispersed over the course of MDV science.

Our analysis of the detrended, normalized distance to nearest field camp data (Fig 5c) revealed three changepoints, 1968, 1974 and 1989, and the associated Mann-Kendall tests identified monotonic trends with Sen's slopes of -0.04, -0.06, and 0.04, respectively. No significant trend appeared after 1989. The first trend is negative, comprising the first decade of MDV research (1958–1968). We attribute the changepoint in 1968 to the construction of four field camps that year, which decreased the randomized camp distance by approximately 16 km (21%, Fig 5c). The second period (1968–1974) continues the negative trend. Over the combined period (1958–1974), the normalized distance between study sites and field camps significantly decreased, which suggests research was being conducted closer to field camps independent of the new camps making study sites more accessible. At the same time, the percent area of the standard distance circle outside the field camp bounding polygon decreased by 5% per year during the same combined period (Fig 5a). This furthers the notion that the spatial extent of study sites was contracting as research was becoming more localized within regions serviced by field camps. The third period (1974–1989) shows a positive trend of normalized

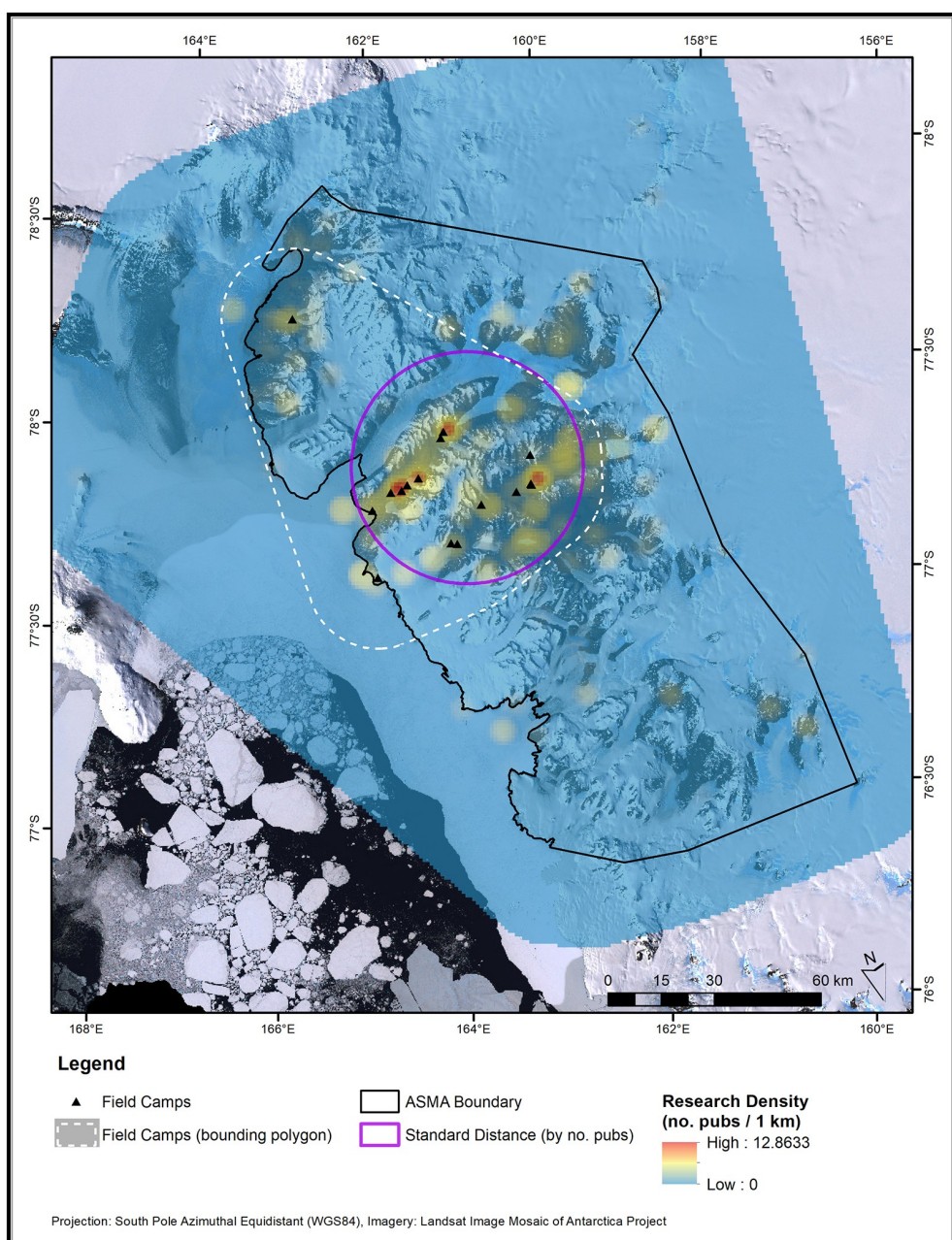

**Fig 4. Hotspots (density), and standard distance (dispersion) of scientific field studies in the McMurdo Dry Valleys from 1907–2016.** The minimum bounding polygon for all historic and current field camps (with 20 km buffer), and the McMurdo Dry Valleys Antarctic Specialty Managed Area (ASMA) boundary are included for reference. Note that the density surface is displayed with a stretch (gamma = 4) to highlight mid-range values. An animation showing these data through time is available in the S1 Video.

distance between field camps and study sites, indicating more dispersed research (Fig 5c). The most recent 28 years of research (1989–2016) showed no significant trend in normalized distance.

For study sites within a 5 km radius of each field camp, 64% (1,779/2,778) occurred while the camp was in place. Similarly, for study sites within 20 km, 61% (6,876/11,358) occurred

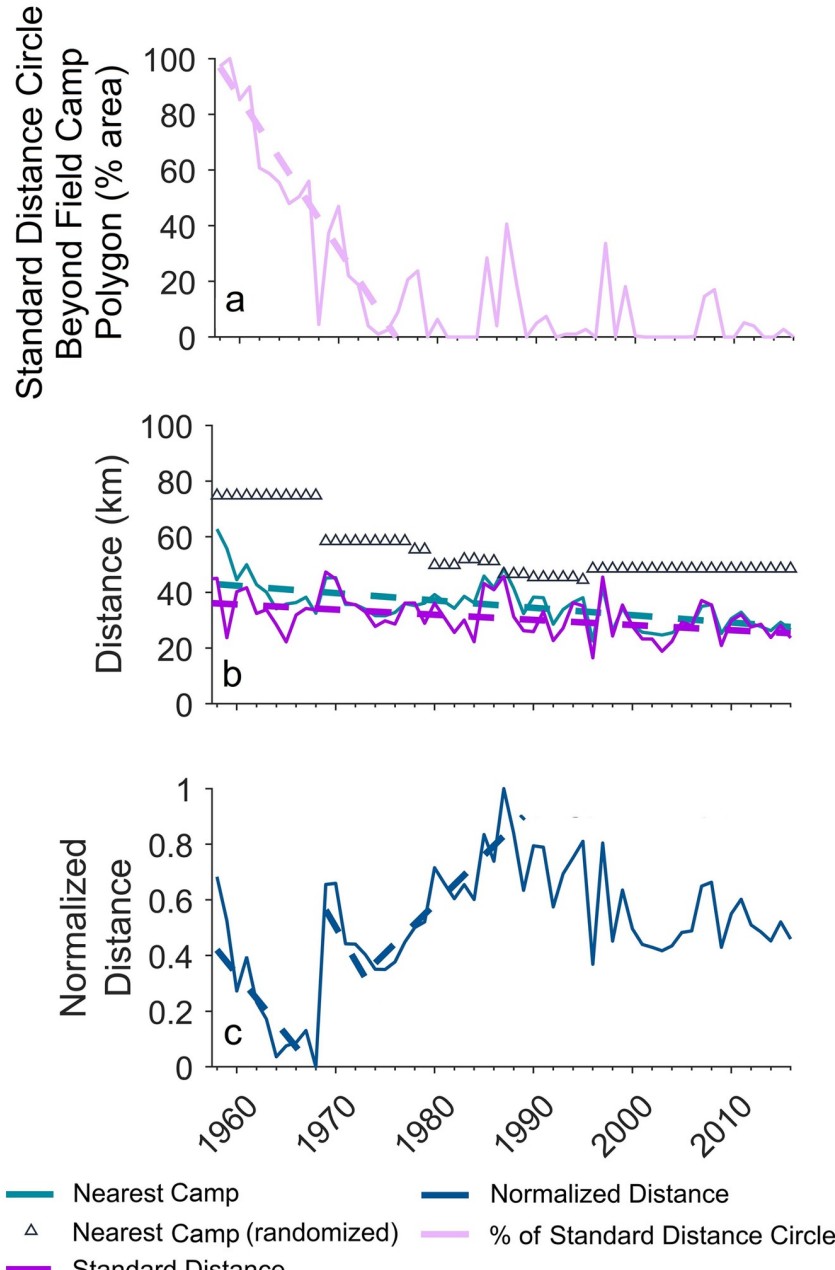

**Fig 5. Time series of field camp proximity and overlay analyses.** Only statistically significant (p < 0.05) Sen's slopes (dashed lines) are shown. a) Percent area of study site standard distance circle outside of field camp bounding polygon; b) Distance to the nearest field camp, standard distance of study sites, and randomized distance to the nearest field camp; c) Distance to the nearest field camp detrended from randomized distance and normalized.

while the field camp was in place. At both the 5 km radius and the 20 km radius a single publication was frequently associated with multiple camps and therefore counted multiple times. At the 20 km radius this overlap was an order of magnitude greater than the actual number of publications. Across all camps, at both < 5 km and < 20 km distances, over 60% of research was published while camps were in place, although this differed among individual camps (Fig 6).

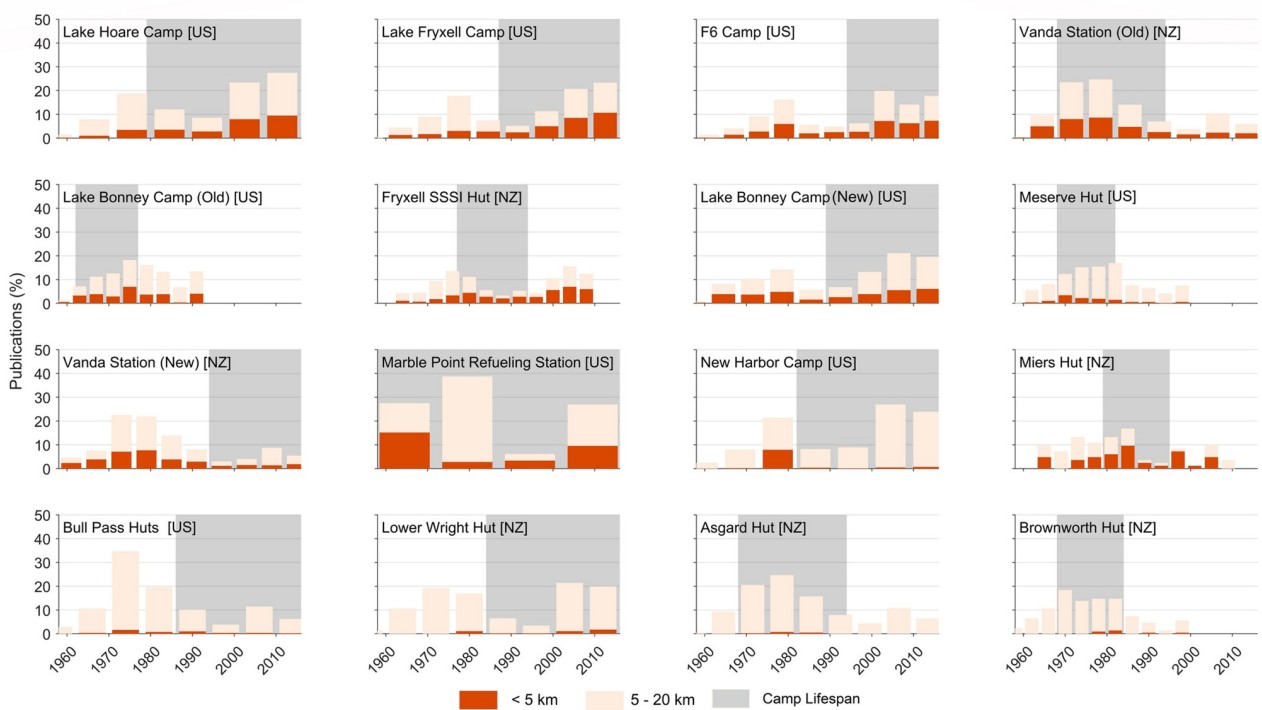

**Fig 6. Percentage of publications within 5 km and 5–20 km of each field camp, relative to the total number of publications within 20 km of the camp from 1958–2016.** Gray shading indicates the lifespan of each camp and bar widths correspond to quartiles based on this lifespan. Field camps are sequentially ordered (left to right) based on how many publications were counted within 5 km of them during their lifespan. Field camps managed by New Zealand (NZ) and the United States (US) are indicated in brackets.

For study sites located within 5 km of a field camp, we compared the number of publications in the quartile immediately preceding and immediately following the construction of each camp. Results ranged from a change of +27 [0 to 27] (Marble Point Refueling Station) to -74 [78 to 4] (New Harbor Camp), with an average change of +1.3. This suggests that the immediate effect of camp construction on publication output was close to negligible. When we extended the comparison to two quartiles before and two quartiles after camp construction, results ranged from +94 [43 to 137] (Vanda Station [Old]) to -75 [80 to 5] (New Harbor Camp). The average change in publications was +14.1, more than ten times the average increase in the first quartile alone. This suggests that field camp construction has a positive effect on local scientific productivity, and the delay is likely due to the time between data collection and publication (see Tables A and B in S1 Dataset for detailed results).

By the end of 2016, seven (44%) of the 16 total field camps in the MDV had been removed. Comparing publications in the final quartile of camp existence with the quartile immediately following its removal, results ranged from -17 [36 to 19] (Lake Bonney Camp (Old)) to +4 [2 to 6] (Miers Hut), with an average of -5.7. Comparing two quartiles immediately before and after camp removal, results ranged from -29 [61 to 32] (Vanda Station (Old)) to +23 [57 to 80] (Fryxell SSSI Camp), with an average change of -5.4.

## Discussion

The clustering of field camps and research sites in Taylor and Wright Valleys has no impact on our changepoint analysis or proximity analysis, but it does create overlapping research spheres that often inflated our publication counts for individual field camps. This impacted our ability

to precisely determine the effect that field camp construction and removal had on research in the surrounding area. For example, the publications that followed the construction of the new Vanda Station camp in Wright Valley cannot be compared to a period when no camp existed because the recently removed old Vanda Station was in a nearby location. Similar overlaps occur in Taylor Valley, especially around Lake Fryxell. While the overall results of the camp-by-camp analysis conform to our expectation that research in the area surrounding a field camp would increase in the years a camp was present and decrease following its removal, the results from individual camps do sometimes contradict our expectations. It is clear from our results that the construction of a field camp does not inevitably lead to an increase in research in the surrounding area, as demonstrated by examples such as the Miers Hut field camp. The camps that do not conform to our expectations tend to be located in more peripheral areas of the MDV, in contrast to the highly productive cluster of field camps in Taylor Valley.

Broader historical research on MDV science can help to interpret our quantitative results. While the nature of historical research makes it difficult to attribute direct causation, such contextualization is particularly important for drawing environmental management lessons from our results. We provide such context in the following sections, drawing on historical archives of the MDV [19] and our collective field experiences in the region.

The three periods identified in our analysis of publication frequency versus camp availability correspond to patterns in the broader history of science in the MDV. For example, the first period of simultaneous increases in productivity and field camps is likely a result of the International Geophysical Year (1957–1958) and the commitment of the US and NZ to research programs in Antarctica [51, 52]. The geology-focused Dry Valleys Drilling Project followed as the first 'big science' project in the MDV, and its completion precedes the decline in publications [53]. The scientific focus shifted from geologic research through the 1980s to ecological research in the 1990s [4]. This also marks the intensification of hypothesis-driven research and the beginning of environmental monitoring, exemplified by the work of the MDV Long-Term Ecological Research Project [25]. These changes, in addition to new technologies like remote sensing and rapid DNA sequencing likely contributed to the continued increase in publications despite the flatlining of field camps in the mid-1990s.

Another important consideration is logistical support. Logistics, particularly helicopters, play an important role in MDV field science, influencing what kind of research is possible and where it takes place [36]. The amount of logistical support in the MDV has changed over time and relates to changes in funding and interests of national research programs. Interestingly, our results suggest that access to helicopters has not necessarily meant an increase in the distance between field camps and study sites. A key point is that helicopters are not kept out in the field but fly the 100 km back to McMurdo Station each night. Researchers must therefore plan around this pattern, which may have contributed to the concentration of field camps (and research) within Taylor Valley. Having high densities of field sites and camps in a single valley means helicopters can move in fairly linear flight paths. This makes efficient use of fuel and flight time while shuttling researchers up and down the valley. The consistent access to helicopters enables researchers to coordinate with each other in the field, moving equipment and personnel in response to changing conditions at research sites scattered between field camps. This may incentivize keeping research confined to a smaller area, and likely contributes to the decrease in distance and dispersal over time that we observed. An additional logistical feature that likely has substantial impact on scientific productivity is camp staffing. Having a dedicated cook, mechanic, manager, and other support personnel gives scientists more time to conduct their field research. Integrating demographic information into a measure of camp size (e.g., annual number of people) as well as data on funding (e.g., number of funded research projects; dollar amounts of grants), could serve as a way to weight the publication count for

each camp and refine future assessments. Such data may be difficult to acquire and harmonize and would thus require close collaboration among international programs.

Changes to environmental management policy in the MDV may also be playing a role in reducing the distance between study sites and field camps. Research focused on the environmental impact of science and tourism in the MDV is often followed by policies to protect the environment [54]. In some cases, policies may guide study site selection based on minimizing ecological disturbance while still enabling research questions of the proposed project to be addressed. For example, the area where field camps are centrally located is already disturbed [55, 56] and therefore may often be considered by managers as a good candidate for minimizing overall environmental impact to the MDV.

Some of the variability we observe in terms of how much the construction or removal of a field camp impacts the study of the local area can be attributed to the unique circumstances of each camp [36]. For example, the surprising drop in publications in the New Harbor area that followed the construction of the New Harbor Camp can be attributed to the fact that New Harbor was an important location for the Dry Valleys Drilling Project in the mid-1970s, and that the Dry Valleys Drilling Project happened to end just as the field camp was established. Moreover, the camp was primarily intended to facilitate marine research in McMurdo Sound, the publications from which tend not to be included in our bibliographic data, which focused on terrestrial research. It is important to remember that decisions on field camp placement have not always been made with the goal of maximizing scientific productivity, and overall levels of scientific activity have varied over time.

It is also important to acknowledge that publication frequency is not a perfect measure of scientific productivity. Much early research in the MDV was exploratory and published in small national journals or as 'grey' literature. During this period, there were also instances where similar papers were published by the same authors in multiple venues, as a way to reach a broader audience. In recent decades, authors often compile data from multiple years into one to two papers published in high impact journals. Different types of science require different lengths of time. On one hand, primarily descriptive studies of specific landforms usually take less time to move from the field to the lab to publication than multi-year *in-situ* experiments comparing different types of data from multiple sites. On the other hand, once groups like the MDV Long-Term Ecological Research Project have installed their equipment, it may be possible to produce several high-quality studies in quick succession. Assessing the quality of publications is a complex bibliometric challenge, especially when comparing across disciplines, time periods, and national publishing cultures. Still, future work might be able to tease out the influence of these and other factors on camp productivity by weighting publication frequency with data on article-level metrics (e.g., number of reads, citations, and so on), which are becoming increasingly available in standardized formats. These bibliometric techniques and data should be used critically, however, and complemented by close reading and contextualization [57].

Throughout the period of sustained scientific research in the MDV, it is impossible to make a clear statement about causation; we cannot confidently say whether field camp placement drives scientific research or whether scientific research drives field camp placement. Science and logistics are so intertwined in the MDV that trying to untangle them quickly becomes meaningless. This close connection is an important finding in itself. Nevertheless, more in-depth historical research might productively examine the causes of the patterns and trends revealed in this study, especially those related to international politics. As noted above, it is possible that national disparities in logistical capabilities, or different national policies might help to explain some of our results. Since a trend that applies to one country may not

necessarily apply to another, such information might further help environmental managers to make informed decisions about field camp construction.

## Conclusions

Our analysis of field science in the McMurdo Dry Valleys shows that, on average, study sites have become closer to field camps over time as both field camps and study sites have become centralized in Taylor and Wright Valleys. This finding has important policy implications. On the one hand, an increasing alignment between field camp placement and scientific research presumably makes science more efficient and reduces unnecessary environmental impact. On the other hand, the intensification of research around certain high-productivity camps potentially concentrates environmental disturbance in these areas, which may be detrimental if sustained, focused human activity has more severe and long-lasting impacts on the environment than diffuse activity, as some studies suggest [55, 58]. These considerations should inform assessments of the tradeoffs between environmental impact and scientific output.

A clear implication of our study for environmental managers is that it cannot be assumed that the construction of a field camp will lead to an increase in scientific research in the surrounding area, or that the removal of a field camp will lead to a decrease in research. Given the known environmental impact of field camps, this emphasizes the need for environmental managers to ensure that funding and logistics are in place for sustained scientific research, as well as to articulate a clear scientific rationale prior to the construction of a field camp. Our study shows that the most productive field camps tend to be clustered together, which suggests that particular care needs to be taken when constructing a new field camp in an area without any existing field camps. This is particularly the case where little or no scientific research is already taking place in the area where the field camp is to be constructed. However, our findings also show that, on average, scientific productivity in the local area around a field camp does not significantly increase until the second quartile of a camp's existence. This suggests that once a field camp has been constructed, it should be given time to become productive before a decision is made to remove it.

Although labor-intensive, the approach we developed could be adapted to study the relationship between field camps and scientific research in other parts of Antarctica, and potentially other parts of the world. While the MDV offer an ideal case study for research of this sort, there is nothing that would preclude a similar approach being taken in any region where a sufficient number of publications could be reliably assigned a geographic location, and where study site locations could be associated with field camps. More research on the influence of field camp placement on environmental science would show whether the patterns we observe in the MDV are broadly applicable, or if they are unique to the region's specific geographic and historical circumstances. For example, author and publisher nationality data could shed light on the political dimensions of Antarctic research and provide insights into drivers of international collaboration and competition. Future research could also examine the spatial patterns of different scientific disciplines and address whether field camps serve as sites of interdisciplinarity and cross-pollination. Such analyses may help environmental managers make more nuanced decisions about field camp placement, as what works for one type of science may not work for another.

## Supporting information

**S1 File. Kernel density estimation parameterization.** This describes the process and parameters used to create the heat maps (Fig 4 and S1 Video).
(DOCX)

**S2 File. PRISMA flow diagram.** Details on the review and screening process for bibliographic references.
(DOCX)

**S1 Dataset. Bibliographic and results data.** Bibliographic data; camp distance frequency tables and statistical tests for the proximity analysis (Figs 3 and 5); Table A-Location analysis/quartile counts and Table B-Camp rankings by publication (Fig 6).
(ZIP)

**S2 Dataset. Geospatial data.** Annual GIS layers of camp locations; study site locations with associated publication counts; randomly generated point locations; geospatial layers resulting from analyses of study sites (Figs 2 and 4 and S1 Video).
(ZIP)

**S1 Video. Animated version of Fig 4.** Hotspots (density), and standard distance (dispersion) of scientific field studies in the McMurdo Dry Valleys shown sequentially for each year from 1957–2016. The minimum bounding polygon for all historic and current field camps (with 20 km buffer), and the McMurdo Dry Valleys Antarctic Specialty Managed Area (ASMA) boundary are included for reference. Note that the density surface is displayed with a stretch (gamma = 4) in order to highlight mid-range values.
(AVI)

## Acknowledgments

We appreciate the three anonymous reviewers whose comments significantly improved the final manuscript. We also appreciate the insightful feedback of researchers from the McMurdo Dry Valleys Long-Term Ecological Research Project, particularly Peter Doran. We are also grateful to Bryce Glenn for assistance in identifying study site locations from the research articles, and for geospatial support provided by the Polar Geospatial Center.

## Author Contributions

**Conceptualization:** Stephen M. Chignell, Madeline E. Myers, Adrian Howkins, Andrew G. Fountain.

**Data curation:** Stephen M. Chignell, Madeline E. Myers.

**Formal analysis:** Stephen M. Chignell, Madeline E. Myers, Adrian Howkins.

**Funding acquisition:** Adrian Howkins, Andrew G. Fountain.

**Investigation:** Stephen M. Chignell, Madeline E. Myers, Adrian Howkins.

**Methodology:** Stephen M. Chignell, Madeline E. Myers, Adrian Howkins.

**Project administration:** Stephen M. Chignell, Adrian Howkins, Andrew G. Fountain.

**Resources:** Adrian Howkins, Andrew G. Fountain.

**Software:** Stephen M. Chignell, Madeline E. Myers.

**Supervision:** Adrian Howkins, Andrew G. Fountain.

**Validation:** Stephen M. Chignell, Madeline E. Myers.

**Visualization:** Stephen M. Chignell, Madeline E. Myers.

**Writing – original draft:** Stephen M. Chignell, Madeline E. Myers, Adrian Howkins.

**Writing – review & editing:** Stephen M. Chignell, Madeline E. Myers, Adrian Howkins, Andrew G. Fountain.

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
