## [Decision Letter · Decision Letter 0]

22 Feb 2021

PONE-D-21-01330

Research sites get closer to field camps over time: informing environmental management through a geospatial analysis of science in the McMurdo Dry Valleys, Antarctica

PLOS ONE

Dear Dr. Chignell,

Thank you for submitting your manuscript to PLOS ONE. After careful consideration, we feel that it has merit but does not fully meet PLOS ONE’s publication criteria as it currently stands. Therefore, we invite you to submit a revised version of the manuscript that addresses the points raised during the review process.

Three reviewers have evaluated your manuscript and all have noted the merits of your study. After reading through your manuscript, I agree with their assessment. That said, reviewers have also provided constructive feedback which I believe will be useful as you craft a revision to your manuscript (if you decide to do so). In particular, please note the comment by Reviewer 2 regarding Cochrane systematic review requirements. While your paper does fall short of those criteria, it does provide useful information which, as the reviewer notes, has been included in other papers in this journal. Additional text can be added for clarity on this topic in your revised submission (for example, by noting this in the text of a revised discussion section). Additional concerns that should be addressed include the quality of the research (especially since the numbers of journals have increased over time) and size of field camps.

We look forward to receiving your revised manuscript.

Kind regards,

Charles William Martin

Academic Editor

PLOS ONE

3. We note that Figures 2 and 4 in your submission contain [map/satellite] images which may be copyrighted. All PLOS content is published under the Creative Commons Attribution License (CC BY 4.0), which means that the manuscript, images, and Supporting Information files will be freely available online, and any third party is permitted to access, download, copy, distribute, and use these materials in any way, even commercially, with proper attribution. For these reasons, we cannot publish previously copyrighted maps or satellite images created using proprietary data, such as Google software (Google Maps, Street View, and Earth). For more information, see our copyright guidelines: http://journals.plos.org/plosone/s/licenses-and-copyright.

(1) You may seek permission from the original copyright holder of Figure(s) [#] to publish the content specifically under the CC BY 4.0 license. 

Reviewers' comments:

Reviewer's Responses to Questions

**Comments to the Author**

1. Is the manuscript technically sound, and do the data support the conclusions?

Reviewer #1: Yes

Reviewer #2: Partly

Reviewer #3: Yes

2. Has the statistical analysis been performed appropriately and rigorously? 

Reviewer #1: Yes

Reviewer #2: Yes

Reviewer #3: Yes

3. Have the authors made all data underlying the findings in their manuscript fully available?

Reviewer #1: Yes

Reviewer #2: Yes

Reviewer #3: Yes

4. Is the manuscript presented in an intelligible fashion and written in standard English?

Reviewer #1: Yes

Reviewer #2: Yes

Reviewer #3: Yes

5. Review Comments to the Author

Reviewer #1: A generally well written and interesting paper on the relationship between established field camps and published papers.

Introduction is a bit scanty. Please provide more rationale for hypothesis 1 - background on why distance between study sites and camps would have changed over time?

49 …the environmental impact of human…

71-80 quality and impact as well as quantity

91 … the MDV has been an…

MDV is singular everywhere else in the paragraph – either change as indicated here, or (preferable) correct all the other instances to reflect that “Valleys” are plural.

97-105 Rewrite paragraph. What motivates hypothesis 1? Some reason for testing this is needed.

99 … two hypotheses. First, study…

100-102 Remove “The logistical capacity of a field camp supports more complex, instrument-intensive studies and becomes a draw for new investigators and new research projects.” This provides justification for hypothesis 2 but does not fit in a list of two hypotheses.

103 … established and decreases following…

103 “Finally,” does not make sense in a list of two hypotheses that have already been presented.

Figure 1 provides no useful information and should be removed.

M&M

116-117 What were the search terms used?

116-127 The numbers don't add up. E.g. if there were 1,485 from the NZ bibliography, 1,257 from the Web of Science, and 83 were duplicates, the total number used in the study would be 2,659, not 2,742. Likewise, 664+625+198 does not equal 1490. Simple arithmetic errors like this do not make me worry about the more complex analyses which I am accepting on faith in the authors.

127 … ‘Commonwealth Stream’, etc.), defining 289 unique study…

Figure 2. The inset covers too small an area, please include a larger area to help locate MDV on the Antarctic continent. Latitude and longitude are missing in both inset and main map. Label the seaice/ocean for readers unfamiliar with the area.

164-172 I am unfamiliar with changepoint analysis and unable to assess its suitability for this analysis.

223-232 With so much overlap between the 5 km circles, why not use a smaller diameter to better test hypothesis 1?

Results

239-244 This belongs in Methods not Results.

Figure 4. What is the area over which the density surface is displayed? It appears to extend well beyond the bounding polygon and buffer.

298-299 I would disagree that strategic field camp placement caused the decrease, since of the few new camps that were emplaced, most were placed very close to existing camps. This suggests that logistic rather than scientific reasons drove site selection, as is mentioned in the Discussion. I interpret the decrease to mean that researchers chose to travel shorter distances from field camps to research sites. I believe this is supported by the overlay analysis that shows an extreme decrease in distances from 1958-1977. Please provide clearer support for your position that field camp placement was strategic.

Discussion could be expanded to include proposing a measure of scientific quality as well as quantity of publications. Scientific output measured by number of publications is limited. Similarly, some measure of camp size (number of people) would be an important addition. In the most simplistic terms, more people can accomplish more science.

392-406 An additional logistic feature that likely has substantial impact on scientific productivity is camp staffing. Having dedicated cook, mechanic, management and other not-directly-science-focused support personnel frees up scientist time to do science.

411 …while still enabling addressing the research…

437-450 Delving more deeply into the funding aspect would be useful. Without sufficient funding (perhaps codified as number of funded research projects, amount of funding, or number of science personnel as mentioned above in reference to camp size) the productivity of each camp is probably less. I’m just suggesting adding this, and scientific quality and camp size factors, to the text of the discussion; I am not suggesting redoing the data or analysis!

Supplements

Table A. …buffer areas. These data are visualized in…

Reviewer #2: This manuscript sets out to test two hypotheses related to a region in Antarctica: if study sites become progressively closer to field camps over time and how the number of publications in a surrounding area relates to the availability of a field camp nearby. The authors use an extensive dataset of literature from the area to collect study site information and then they conducted a number of spatial and temporal analyses to investigate how the data relates to their research hypotheses. The authors relied on Mann-Kendall tests for their hypothesis testing, with other descriptive statistics and historical context to interpret their results and trends. The authors clearly state the caveats associated with their chosen statistical methods and encourage several avenues of future research to further investigate these research topics.

The majority of my line-by-line comments are centered around correcting grammar and punctuation, providing additional context for some choices in their methods, and adding greater citation support for certain statements. I consider these all relatively minor revisions and should not preclude the manuscript’s acceptance by the journal.

Much of the dataset the manuscript uses (i.e., publications and study sites within them) was collected through a review of the literature, which while quite extensive, does not seem to meet the requirements of a Cochrane systematic review based on the methods written (e.g., multiple independent reviewers, a pre-published protocol, explicit reference inclusion criteria) and no PRISMA checklist and flow diagram were included in the submission materials. According to the PLoS ONE website, the journal seems to require these items and a Cohrane methodology with any literature review, which I believe this study qualifies as. However, the journal has published bibliometric studies with varying scales of rigor in data collection methods (e.g., Zhang et al. 2019 https://doi.org/10.1371/journal.pone.0210707; Cebrino and de la Cruz 2020 https://doi.org/10.1371/journal.pone.0242781; Nita et al. 2019 https://journals.plos.org/plosone/article?id=10.1371/journal.pone.0217638), so I am unclear on how strictly the systematic review requirements are applied. I believe this study falls short of a systematic review and should not qualify as one but would be sufficient enough to be published if a Cochrane systematic review is not in fact required by the journal.

I believe the research covered in the manuscript is a novel contribution to the scientific literature on a topic that is relevant to environmental managers and researchers working in Antarctica. Also, while the study focused on one region in Antarctica, I believe the questions asked and the results gained could be more widely relevant and that the methods used in the manuscript could be applied to other regions, as the authors themselves suggest. The scope appears comparable to other articles published in PLoS ONE (e.g., Nita et al. 2019 https://journals.plos.org/plosone/article?id=10.1371/journal.pone.0217638; Kang et al. 2019 https://doi.org/10.1371/journal.pone.0225551). Though the literature review did not seem to meet Cochrane systematic review requirements, the data collection was impressively extensive and the spatial and temporal analyses were well-chosen for the authors’ questions. The authors clearly stated their hypotheses and supported their conclusions through in-depth descriptive and inferential statistical techniques and historical context from the literature. For these reasons, provided a systematic review is not required, I recommend this manuscript be accepted with the revisions requested below.

 

Line 45–46: Please include a citation on how Antarctica is among the most protected environments.

Line 46–50: The sentence does not have verb tense consistency and there is an unnecessary comma after “(commonly known as the Madrid Protocol)”.

Line 57: The clarifying definition here is appreciated, though the phrase “semi-permanent” is fairly well-established and standardized in the literature, so it doesn’t need ‘’ scare quotes around it here.

Line 64: The phrase “human footprint” is even more well-established and standardized in the literature, so it doesn’t need ‘’ scare quotes around it either.

Line 74–80: I would appreciate the inclusion of citations here to support the practice of measuring scientific productivity in the way described as well as some of difficulties associated with it. They would not have to be specific to measuring scientific productivity in Antarctica.

Line 83–83: A phrase is written as “ice-free area” and “ice free area” in the same sentence and should be kept consistent throughout this sentence.

Lines 81–96: A lot of information is covered in this paragraph, but only two citations are included to support the size of the area, the ice-free area, and the broader history of environmental management protocols. The literature describing the landscape and history of the region is extensive, at least some of which should be cited here.

Lines 100–102: I don’t really understand either how this sentence connects to the first hypothesis or if it is supposed to be its own hypothesis. I don’t think it needs to be included in the paragraph.

Line 103: The word “decrease” should be “decreases.”

Lines 116–117: I would appreciate more detail in how Web of Science was queried, particularly the inclusion of a search string or terms if any were used as well as the years assessed (if it was just the years after the bibliography or also the years covered by the bibliography). This information is necessary to make the study reproducible.

Lines 123–125: It seems like the math here doesn’t add up. 1,490-664-625=201, not 198. Either the numbers are incorrect or I misunderstood the sentence and it should be clarified.

Lines 132–135: I understand these choices to mark non-point features were necessitated by the decision to use point locations in your analysis and the choices made seem reasonable, but is there any additional information that could be added here clarifying how large the lakes, streams, and glaciers are to account for how far off the point selections could be from the actual study area or proximity to the field camps? How long do the streams tend to be in kilometers or how large are the lakes and glaciers? Some additional context would be reassuring.

Line 143: The dash between years should be an – en dash, and any other ranges in years (e.g., line 337) or other measurements (e.g., line 149) should also use an – en dash.

Lines 145–151: I understand the assigned distances are subjective, but is there any additional justification for their selection that could be added, either based on travel times around the region or expert knowledge of working in the area?

Lines 233: Change “with the period” to “with the periods” to match the plurality of the initial use of the word.

Lines 239–244: This paragraph seems primarily to serve as justification for inclusion or exclusion of certain data points in different analyses and might fit better in the methods.

Line 284: Fig 5c is mentioned here, but Fig 5a isn’t mentioned until line 298, so perhaps the order in the figure itself should be changed.

Line 294: I believe this is meant to be a subheading for the results? It’s hard to tell because the same font style was used as the results heading. If it’s supposed to be its own main section, I think it all belongs under results instead. If it is in fact a subsection, then I think it’s unnecessary or at least other subheadings should be used so it’s not the only subsection in the results.

Line 320: This figure seems to introduce the term “hut” for the first time. Is “hut” meant to be the same as “camp”? If so, the term should be standardized as “camp” in the figure legend, or if not, a definition of “hut” vs. “camp” should be included in the caption and potentially elsewhere in the paper.

Line 328–329: This sentence confused me. I think the purpose was to point out that there were more study sites than publications, but I’m not really sure and some rewording would help. The use of the word “overlap” here could perhaps be explained a bit better as well.

Lines 343–350: What is the difference in the comparisons included here? One is for the quartiles “immediately preceding and immediately following the construction of each camp” and the second is “the two quartiles before and after field camp construction,” which seem like they could be the same thing. My guess would be that the first compared one quartile with another quartile and the second compared two quartiles with two quartiles, but it is pretty confusing to read as is.

Lines 350–352: This seems to be an important caveat to your comparisons of publication year and years that a camp was in use. I believe the difference between the year that the research was conducted and the year that the research was published could be brought up here as well as earlier (perhaps in the methods bibliometric data section with a sentence specifying publication year data was collected instead of study year data) since this data is such an important component of your results and conclusions.

Line 411: The phrase “enabling able to” should be corrected.

Line 426: This conclusions section seems very long and that many of topics covered, such as the point about national disparities and the other future research recommendations, may fit better in the discussion section.

Line 435: More citations should be added to support the use of the phrase “some studies” or the phrase should be changed to reflect the fact that only one study is cited.

Line 454: The word choice “quickly” does not seem to fit the sentence.

Reviewer #3: This is a fantastic paper about the changes in location of research projects relative to the location of field camps in the McMurdo Dry Valleys, Antarctica. It is written very well and follows a clear and obvious direction. The video in the supplementary material is a wonderful addition.

The only largest suggestion I have is relatively minor. I feel that it is important to include a few sentences in the discussion stating the caveat that publications aren't necessarily indicative of the amount or quality of research being conducted. Although this is a really great study, I don't think it is wise to conclude that the amount of research is fully represented by the number of publications.

I have a few minor comments.

It would be nice to add some p-values to Figure 3 (caption or figure) or related test for the growing body of scientists who no longer believe in statistical significance. It would also be nice to add the p values to other parts of the text that mentions significance, e.g., the paragraph starting at line 302. Other reviewers or the editor might prefer these significance values to be included in the supplementary material but I would prefer them to be included in the text.

line 269 and 360: Capitalize 'Valleys'

6. PLOS authors have the option to publish the peer review history of their article (what does this mean?). If published, this will include your full peer review and any attached files.

Reviewer #1: No

Reviewer #2: No

Reviewer #3: No

---

## [Author Response · Author response to Decision Letter 0]

21 Aug 2021

Response to Reviewers

We sincerely appreciate the excellent comments on our first submission. We have responded to each reviewer concern below.

Reviewer #1: A generally well written and interesting paper on the relationship between established field camps and published papers.

Introduction is a bit scanty. Please provide more rationale for hypothesis 1 - background on why distance between study sites and camps would have changed over time?

We have reorganized and added text and citations to the Introduction, including additional justification for our hypotheses. The section leading up to hypothesis 1 now reads: 

“The ebb and flow of established field camps and scientific activity offers an opportunity to examine how research activity in the region responds to camp establishment and removal. Given the challenging environmental conditions and the desire of researchers and national research programs to make efficient use of time and resources, it seems likely that the construction of a new field camp would encourage researchers to study areas nearby the camp. This might be spurred further by the need to reduce environmental impact following the adoption of the Madrid Protocol in 1991. It also seems likely that the increased logistical capacity provided by a field camp would support more complex, instrument-intensive studies and become a draw for new investigators and new research projects. In this way, field camps might act as centers of gravity that ‘pull’ researchers (and their study sites) toward them over time. Following this line of reasoning, one would also expect this attractive force to disappear after the removal of the field camp and researchers’ attention to shift toward other regions.” 

We think this provides adequate background for the hypotheses that follow, without going into too much detail about factors (e.g. helicopters; funding) that are addressed later in the Discussion. 

49 …the environmental impact of human…

This has been corrected.

71-80 quality and impact as well as quantity

This section has been updated with new language and citations: “In addition to journal rankings and citation metrics, scientific productivity is often measured by the number of research publications [16–18]. However, assigning specific locations to the research described in each publication can be challenging and time-consuming.”

91 … the MDV has been an…

This has been corrected.

MDV is singular everywhere else in the paragraph – either change as indicated here, or (preferable) correct all the other instances to reflect that “Valleys” are plural.

We agree with your suggestion to make MDV plural and have edited the text throughout to reflect this.

97-105 Rewrite paragraph. What motivates hypothesis 1? Some reason for testing this is needed.

We have rewritten this paragraph as suggested, focusing it on our hypotheses and approach. We have included additional background and reasoning to the preceding paragraph for clarity.

99 … two hypotheses. First, study…

This has been corrected.

100-102 Remove “The logistical capacity of a field camp supports more complex, instrument-intensive studies and becomes a draw for new investigators and new research projects.” This provides justification for hypothesis 2 but does not fit in a list of two hypotheses.

Thank you for noting this. We have moved this to the preceding paragraph.

103 … established and decreases following…

This has been corrected.

103 “Finally,” does not make sense in a list of two hypotheses that have already been presented.

We have edited the sentence so it now reads: “We then consider the implications of our results for environmental management…”

Figure 1 provides no useful information and should be removed.

We respectfully disagree, and think that the photograph provides valuable context of a typical field camp as well as the MDV landscape, especially for readers unfamiliar with Antarctic/MDV field science.

116-117 What were the search terms used?

We have provided the Web of Science query below and inserted it into the Methods section of the manuscript. ‘TS’ stands for the ‘Topic’ field tag, which searches title, abstract, and keywords of all indexed references in the Web of Science. 

(TS=("McMurdo Dry Valleys") OR TS=("Taylor Dry Valley") OR TS=("Wright Dry Valley") OR TS=("Victoria Dry Valley") OR TS=("Taylor Valley") OR TS=("Wright Valley") OR TS=("Victoria Valley") OR TS=(Dry Valley*) OR TS=(Ice-free Valley*)) AND TS=(Antarctica)

116-127 The numbers don't add up. E.g. if there were 1,485 from the NZ bibliography, 1,257 from the Web of Science, and 83 were duplicates, the total number used in the study would be 2,659, not 2,742. 

Likewise, 664+625+198 does not equal 1490. Simple arithmetic errors like this do not make me worry about the more complex analyses which I am accepting on faith in the authors.

Thank you for identifying these errors. We have reviewed the bibliographic data and realized we had miscounted and reported the wrong numbers in the first draft. We have since updated the text and all relevant figures with the correct counts. We also rewrote this section to make the language more precise and created a new supplementary figure (PRISMA flow diagram) which shows each step in the bibliographic data collection and screening workflow.

127 … ‘Commonwealth Stream’, etc.), defining 289 unique study…

We have updated the text to reflect this change in phrasing.

Figure 2. The inset covers too small an area, please include a larger area to help locate MDV on the Antarctic continent. Latitude and longitude are missing in both inset and main map. Label the seaice/ocean for readers unfamiliar with the area.

We have redesigned this figure. For the main map, we have added a label for “McMurdo Sound” as well as latitude and longitude markers (we also added the latter to Fig 4). We have moved the inset map and zoomed out the extent to show the wider region of the MDV. We have also redesigned the locator map to show the Antarctic continent in more detail and added extent frames with arrows connecting to each map. We did not add latitude and longitude to the inset map because this information is now available in detail on the main map and because we wanted to provide space to include additional information on the data sources and projection used.

164-172 I am unfamiliar with changepoint analysis and unable to assess its suitability for this analysis.

We have added additional description of the changepoint analysis technique to the text. 

223-232 With so much overlap between the 5 km circles, why not use a smaller diameter to better test hypothesis 1?

We actually did include a smaller diameter circle (1 km), as well as a larger diameter circle (10-20 km) as part of the original analysis. However, we omitted these because we felt that the 1 km results did not add enough important information beyond what was captured in the 5 km results, and unnecessarily complicated Fig 6 (see earlier version of Fig 6 below). 

Results

239-244 This belongs in Methods not Results.

We agree and have updated the text to reflect this.

Figure 4. What is the area over which the density surface is displayed? It appears to extend well beyond the bounding polygon and buffer.

This area is the minimum bounding polygon (with buffer) for all study sites over all years analyzed. The extent appears large because it is based off the cumulative dataset, which includes several studies that took place beyond the MDV ASMA boundary (these appear as small yellow areas within the blue density surface).

298-299 I would disagree that strategic field camp placement caused the decrease, since of the few new camps that were emplaced, most were placed very close to existing camps. This suggests that logistic rather than scientific reasons drove site selection, as is mentioned in the Discussion. I interpret the decrease to mean that researchers chose to travel shorter distances from field camps to research sites. I believe this is supported by the overlay analysis that shows an extreme decrease in distances from 1958-1977. Please provide clearer support for your position that field camp placement was strategic.

The reviewer’s disagreement appears to the result of differing interpretations of the term “strategic”. We share the reviewer’s interpretation of the data and were trying to communicate the same ideas in this passage. In order to clarify this in the text, we have changed the sentence to “logistic rather than scientific reasons drove site selection”, as the reviewer suggests.

Discussion could be expanded to include proposing a measure of scientific quality as well as quantity of publications. Scientific output measured by number of publications is limited. Similarly, some measure of camp size (number of people) would be an important addition. In the most simplistic terms, more people can accomplish more science.

We agree. In the Introduction, we have added references to papers that use publications counts as a measure of scientific productivity, including in the Antarctic context, as well as mention of the other factors affecting productivity: 

“In addition to journal rankings and citation metrics, scientific productivity is often measured by the number of research publications [20–22]. While there are many other factors influencing the productivity of specific camps (e.g., funding, personnel, remoteness), publication frequency provides a consistent metric that can be computed and compared across the full history of scientific activity.” 

We have also expanded the Discussion to include additional factors: 

“It is also important to acknowledge that publication frequency is not a perfect measure of scientific productivity. Much early research in the MDV was exploratory and published in small national journals or as ‘grey’ literature. In recent decades, authors often compile data from multiple years into one to two papers published in high impact journals. Different types of science require different lengths of time. On one hand, primarily descriptive studies of specific landforms usually take less time to move from the field to the lab to publication than multi-year in-situ experiments comparing different types of data from multiple sites. On the other hand, once groups like the Long-Term Ecological Research Project have installed their equipment, it may be possible to produce a several high-quality studies in quick succession. Assessing the quality of publications is a complex bibliometric challenge, especially when comparing across disciplines, time periods, and national publishing cultures. Still, future work might be able to tease out the influence of these and other factors on camp productivity by weighting publication frequency with data on article-level metrics (e.g., number of reads, citations, and so on), which are becoming increasingly available in standardized formats. These bibliometric techniques and data should be used critically, however, and complemented by close reading and contextualization [57].”

Additionally, see our response your subsequent questions below.

392-406 An additional logistic feature that likely has substantial impact on scientific productivity is camp staffing. Having dedicated cook, mechanic, management and other not-directly-science-focused support personnel frees up scientist time to do science.

We agree and have added the following text to the Discussion:

“An additional logistical feature that likely has substantial impact on scientific productivity is camp staffing. Having a dedicated cook, mechanic, manager, and other support personnel gives scientists more time to conduct their field research. Integrating demographic information into a measure of camp size (e.g., annual number of people) as well as data on funding (e.g., number of funded research projects; dollar amounts of grants), could serve as a way to weight the publication count for each camp and refine future assessments. Such data may be difficult to acquire and harmonize and would thus require close collaboration among international programs.

411 …while still enabling addressing the research…

This has been corrected.

437-450 Delving more deeply into the funding aspect would be useful. Without sufficient funding (perhaps codified as number of funded research projects, amount of funding, or number of science personnel as mentioned above in reference to camp size) the productivity of each camp is probably less. I’m just suggesting adding this, and scientific quality and camp size factors, to the text of the discussion; I am not suggesting redoing the data or analysis!

We have added text on funding and other factors to the Discussion (see answers to previous questions above).

Supplements

Table A. …buffer areas. These data are visualized in…

The captions have been updated.

Reviewer #2: This manuscript sets out to test two hypotheses related to a region in Antarctica: if study sites become progressively closer to field camps over time and how the number of publications in a surrounding area relates to the availability of a field camp nearby. The authors use an extensive dataset of literature from the area to collect study site information and then they conducted a number of spatial and temporal analyses to investigate how the data relates to their research hypotheses. The authors relied on Mann-Kendall tests for their hypothesis testing, with other descriptive statistics and historical context to interpret their results and trends. The authors clearly state the caveats associated with their chosen statistical methods and encourage several avenues of future research to further investigate these research topics.

The majority of my line-by-line comments are centered around correcting grammar and punctuation, providing additional context for some choices in their methods, and adding greater citation support for certain statements. I consider these all relatively minor revisions and should not preclude the manuscript’s acceptance by the journal.

Much of the dataset the manuscript uses (i.e., publications and study sites within them) was collected through a review of the literature, which while quite extensive, does not seem to meet the requirements of a Cochrane systematic review based on the methods written (e.g., multiple independent reviewers, a pre-published protocol, explicit reference inclusion criteria) and no PRISMA checklist and flow diagram were included in the submission materials. According to the PLoS ONE website, the journal seems to require these items and a Cohrane methodology with any literature review, which I believe this study qualifies as. However, the journal has published bibliometric studies with varying scales of rigor in data collection methods (e.g., Zhang et al. 2019 https://doi.org/10.1371/journal.pone.0210707; Cebrino and de la Cruz 2020 https://doi.org/10.1371/journal.pone.0242781; Nita et al. 2019 https://journals.plos.org/plosone/article?id=10.1371/journal.pone.0217638), so I am unclear on how strictly the systematic review requirements are applied. I believe this study falls short of a systematic review and should not qualify as one but would be sufficient enough to be published if a Cochrane systematic review is not in fact required by the journal.

Thank you for this helpful comment. We have reviewed the Cochrane requirements and added the following text to the Methods section:

“While our study does not qualify as a systematic review or meta-analysis according to the Cochrane methodology, we conducted a thorough and extensive analysis combining in-depth descriptive and inferential statistical techniques and historical context from the literature. We have included a modified PRISMA flow diagram detailing our review and screening process.” 

To further clarify our review approach, we have also created a PRISMA flow diagram based on Moher et al. (2009). This is now referenced in the main text and included in the supporting information.

I believe the research covered in the manuscript is a novel contribution to the scientific literature on a topic that is relevant to environmental managers and researchers working in Antarctica. Also, while the study focused on one region in Antarctica, I believe the questions asked and the results gained could be more widely relevant and that the methods used in the manuscript could be applied to other regions, as the authors themselves suggest. The scope appears comparable to other articles published in PLoS ONE (e.g., Nita et al. 2019 https://journals.plos.org/plosone/article?id=10.1371/journal.pone.0217638; Kang et al. 2019 https://doi.org/10.1371/journal.pone.0225551). Though the literature review did not seem to meet Cochrane systematic review requirements, the data collection was impressively extensive and the spatial and temporal analyses were well-chosen for the authors’ questions. The authors clearly stated their hypotheses and supported their conclusions through in-depth descriptive and inferential statistical techniques and historical context from the literature. For these reasons, provided a systematic review is not required, I recommend this manuscript be accepted with the revisions requested below.

Line 45–46: Please include a citation on how Antarctica is among the most protected environments.

We have included a citation “The Greening of Antarctica: Environment, Science and Diplomacy, 1959-1980” Antonnello (2014)

Line 46–50: The sentence does not have verb tense consistency and there is an unnecessary comma after “(commonly known as the Madrid Protocol)”.

We have removed the comma and changed “prohibits” to “prohibited” to be consistent with the past tense verbs in the rest of the sentence.

Line 57: The clarifying definition here is appreciated, though the phrase “semi-permanent” is fairly well-established and standardized in the literature, so it doesn’t need ‘’ scare quotes around it here.

We agree and have removed the scare quotes.

Line 64: The phrase “human footprint” is even more well-established and standardized in the literature, so it doesn’t need ‘’ scare quotes around it either.

We agree and have removed the scare quotes.

Line 74–80: I would appreciate the inclusion of citations here to support the practice of measuring scientific productivity in the way described as well as some of difficulties associated with it. They would not have to be specific to measuring scientific productivity in Antarctica.

We have updated this section to include citations to well-known studies that use publications as a measure of scientific productivity, including a recent one in Antarctica (Jang et al. 2020).

Line 83–83: A phrase is written as “ice-free area” and “ice free area” in the same sentence and should be kept consistent throughout this sentence.

We have changed all cases to “ice-free area”.

Lines 81–96: A lot of information is covered in this paragraph, but only two citations are included to support the size of the area, the ice-free area, and the broader history of environmental management protocols. The literature describing the landscape and history of the region is extensive, at least some of which should be cited here.

We have added 12 additional citations here and added text describing the region’s importance for various scientific questions including as a terrestrial analog for Mars and as a sentinel for global climate change.

Lines 100–102: I don’t really understand either how this sentence connects to the first hypothesis or if it is supposed to be its own hypothesis. I don’t think it needs to be included in the paragraph.

We agree that this was confusing in its original placement between the two hypotheses. We have moved this text to the preceding section with the other contextual information. The hypotheses are now presented clearly in succession.

Line 103: The word “decrease” should be “decreases.”

This has been corrected.

Lines 116–117: I would appreciate more detail in how Web of Science was queried, particularly the inclusion of a search string or terms if any were used as well as the years assessed (if it was just the years after the bibliography or also the years covered by the bibliography). This information is necessary to make the study reproducible.

The complete Web of Science covered the years 1900-2016 and searched the Web of Science Core Collection. The query was as follows:

(TS=("McMurdo Dry Valleys") OR TS=("Taylor Dry Valley") OR TS=("Wright Dry Valley") OR TS=("Victoria Dry Valley") OR TS=("Taylor Valley") OR TS=("Wright Valley") OR TS=("Victoria Valley") OR TS=(Dry Valley*) OR TS=(Ice-free Valley*)) AND TS=(Antarctica)

This is now included in the Methods section.

Lines 123–125: It seems like the math here doesn’t add up. 1,490-664-625=201, not 198. Either the numbers are incorrect or I misunderstood the sentence and it should be clarified.

Thank you for identifying this error. The correct numbers are 1486-628-658=200. This has been updated in the revised manuscript. 

Lines 132–135: I understand these choices to mark non-point features were necessitated by the decision to use point locations in your analysis and the choices made seem reasonable, but is there any additional information that could be added here clarifying how large the lakes, streams, and glaciers are to account for how far off the point selections could be from the actual study area or proximity to the field camps? How long do the streams tend to be in kilometers or how large are the lakes and glaciers? Some additional context would be reassuring.

We have added additional context. The section now reads:

“For linear features such as streams, we used the location of the stream gauge. For ungauged streams or those with multiple gauges, we used the midpoint of the stream feature. For polygons and multipoint features such as lakes and glaciers, we used the centroid of the polygon. These points are approximations, since the actual locations of study sites would vary for each feature and study. For example, field sampling for a study on glacial discharge would tend to take place near the edge of the glacier, whereas a study sampling cryoconite holes would be more centrally located. Based on the MDV Long-Term Ecological Research Project GIS data, the average area of the 130 named glaciers is 46.7 km2, the average area of the 28 named lakes is 1.2 km2, and the average length of the 28 named streams is 3.1 km. The latter calculation excludes the Onyx River, which has a length of 41.6 km. Given that most field sampling of the streams takes place at gauges, and the relatively small size of the lakes, the four or five largest glaciers in the MDV represent the greatest source of uncertainty in our analysis. Although these glaciers lower our precision in some areas, it is reasonable to assume the feature centroids are accurate markers of study locations at the regional scale. Our approach thus provides an imperfect but consistent way of comparing study site locations with uncertain and variable location information across multiple types of features.”

Line 143: The dash between years should be an – en dash, and any other ranges in years (e.g., line 337) or other measurements (e.g., line 149) should also use an – en dash.

Thank you, we have made these changes throughout the text.

Lines 145–151: I understand the assigned distances are subjective, but is there any additional justification for their selection that could be added, either based on travel times around the region or expert knowledge of working in the area?

We have provided addition details justifying the selection of these distances, including reference to supporting archival data (Bromley, A.M. 1975, National Archive of New Zealand). The full citation to the archive is included in the updated reference list and the section in the manuscript now reads: 

“We assumed that research activity within a 5 km radius of a field camp used that camp. We chose this distance based on our own fieldwork experiences in the area and common travel times to and from field study sites. For comprehensive inclusion of all research activity, we assumed that research within a 20 km radius around each camp may have an association with that camp, acknowledging that overlap between camp radii may occur. Based on the historical record, 20 km is the approximate upper limit that someone would travel to a research site (e.g., trips from the old Vanda Station to Don Juan Pond and The Labyrinth).” 

Lines 233: Change “with the period” to “with the periods” to match the plurality of the initial use of the word.

This has been corrected.

Lines 239–244: This paragraph seems primarily to serve as justification for inclusion or exclusion of certain data points in different analyses and might fit better in the methods.

We agree and have moved the paragraph to the Methods section.

Line 284: Fig 5c is mentioned here, but Fig 5a isn’t mentioned until line 298, so perhaps the order in the figure itself should be changed.

We have changed the order of the figure so that the old 5c is now 5a and have updated the relevant mentions in the text.

Line 294: I believe this is meant to be a subheading for the results? It’s hard to tell because the same font style was used as the results heading. If it’s supposed to be its own main section, I think it all belongs under results instead. If it is in fact a subsection, then I think it’s unnecessary or at least other subheadings should be used so it’s not the only subsection in the results.

Yes, this was intended to be a subheading in the results. We agree that it is unnecessary and have removed it.

Line 320: This figure seems to introduce the term “hut” for the first time. Is “hut” meant to be the same as “camp”? If so, the term should be standardized as “camp” in the figure legend, or if not, a definition of “hut” vs. “camp” should be included in the caption and potentially elsewhere in the paper.

Yes, this was a holdover from an earlier version of the manuscript when we were using the term “hut” instead of “camp”. We have updated the figure so that it reads “Camp”. 

Line 328–329: This sentence confused me. I think the purpose was to point out that there were more study sites than publications, but I’m not really sure and some rewording would help. The use of the word “overlap” here could perhaps be explained a bit better as well.

We agree that this was confusing and have revised this section to improve its clarity.

Lines 343–350: What is the difference in the comparisons included here? One is for the quartiles “immediately preceding and immediately following the construction of each camp” and the second is “the two quartiles before and after field camp construction,” which seem like they could be the same thing. My guess would be that the first compared one quartile with another quartile and the second compared two quartiles with two quartiles, but it is pretty confusing to read as is.

Your interpretation is correct, and we have rewritten the sentence to make this clear: 

“For study sites located within 5 km of a field camp, we compared the number of publications in the quartile immediately preceding and immediately following the construction of each camp. Results ranged from a change of +27 [0 to 27] (Marble Point Refueling Station) to -74 [78 to 4] (New Harbor Camp), with an average change of +1.4. This suggests that the immediate effect of camp construction on publication output was close to negligible. When we extended the comparison to two quartiles before and two quartiles after camp construction, results ranged from +94 [43 to 137] (Vanda Station [Old]) to -75 [80 to 5] (New Harbor Camp).” 

Lines 350–352: This seems to be an important caveat to your comparisons of publication year and years that a camp was in use. I believe the difference between the year that the research was conducted and the year that the research was published could be brought up here as well as earlier (perhaps in the methods bibliometric data section with a sentence specifying publication year data was collected instead of study year data) since this data is such an important component of your results and conclusions.

This is an important point, and we have added the following paragraph to the bibliometric data section of the Methods as suggested:

“An important caveat is that we only collected temporal data on the year of publication, not the year researchers conducted the fieldwork. The latter information is not reliably included in published manuscripts, and fieldwork for a single study can take place over more than one season. To minimize this uncertainty and maximize the number of references included in the analysis, we used publication year because it is consistent and available across all references. Still, we recognize the lag between fieldwork and publication and discuss its potential affects in our interpretation of the results.”

Line 411: The phrase “enabling able to” should be corrected.

This has been corrected.

Line 426: This conclusions section seems very long and that many of topics covered, such as the point about national disparities and the other future research recommendations, may fit better in the discussion section.

We agree and, as suggested, have moved the paragraph discussing the intertwined nature of science and logistics, as well as national disparities in logistical capabilities, to the end of the Discussion. We think this rounds out the Discussion section nicely and makes the Conclusion more concise.

Line 435: More citations should be added to support the use of the phrase “some studies” or the phrase should be changed to reflect the fact that only one study is cited.

We have added an additional study, Ayers et al. (2008) to support this statement.

Line 454: The word choice “quickly” does not seem to fit the sentence.

We agree and have removed the word “quickly” from the sentence.

Reviewer #3: This is a fantastic paper about the changes in location of research projects relative to the location of field camps in the McMurdo Dry Valleys, Antarctica. It is written very well and follows a clear and obvious direction. The video in the supplementary material is a wonderful addition.

The only largest suggestion I have is relatively minor. I feel that it is important to include a few sentences in the discussion stating the caveat that publications aren't necessarily indicative of the amount or quality of research being conducted. Although this is a really great study, I don't think it is wise to conclude that the amount of research is fully represented by the number of publications.

We agree and have added text to the Introduction and Discussion that caveats or results (see responses to previous reviewer comments for details). We think the limitations of our approach are adequately described in the revised manuscript.

I have a few minor comments.

It would be nice to add some p-values to Figure 3 (caption or figure) or related test for the growing body of scientists who no longer believe in statistical significance. It would also be nice to add the p values to other parts of the text that mentions significance, e.g., the paragraph starting at line 302. Other reviewers or the editor might prefer these significance values to be included in the supplementary material but I would prefer them to be included in the text.

As suggested, we have added p-values for each slope in the caption of Figure 3. While we understand the reviewer’s preference for including p-values in the text, we would prefer to leave the Results as is for the sake of the flow of the text. We clearly state in the Methods that “significant” always means p < 0.05, in order to direct readers, have added “See S2 Dataset for detailed results of all statistical tests” after the first sentence of the Results section. We think this strikes a good compromise between transparent reporting and readability.

line 269 and 360: Capitalize 'Valleys'

This has been corrected.

---

## [Decision Letter · Decision Letter 1]

15 Sep 2021

Research sites get closer to field camps over time: Informing environmental management through a geospatial analysis of science in the McMurdo Dry Valleys, Antarctica

PONE-D-21-01330R1

Dear Dr. Chignell,

We’re pleased to inform you that your manuscript has been judged scientifically suitable for publication and will be formally accepted for publication once it meets all outstanding technical requirements.

Kind regards,

Charles William Martin

Academic Editor

PLOS ONE

Additional Editor Comments (optional):

Reviewers' comments:

Reviewer's Responses to Questions

**Comments to the Author**

1. If the authors have adequately addressed your comments raised in a previous round of review and you feel that this manuscript is now acceptable for publication, you may indicate that here to bypass the “Comments to the Author” section, enter your conflict of interest statement in the “Confidential to Editor” section, and submit your "Accept" recommendation.

Reviewer #1: All comments have been addressed

Reviewer #2: All comments have been addressed

2. Is the manuscript technically sound, and do the data support the conclusions?

Reviewer #1: Yes

Reviewer #2: (No Response)

3. Has the statistical analysis been performed appropriately and rigorously? 

Reviewer #1: Yes

Reviewer #2: (No Response)

4. Have the authors made all data underlying the findings in their manuscript fully available?

Reviewer #1: Yes

Reviewer #2: (No Response)

5. Is the manuscript presented in an intelligible fashion and written in standard English?

Reviewer #1: Yes

Reviewer #2: (No Response)

6. Review Comments to the Author

Reviewer #1: The authors have done an excellent job of addressing all comments completely. I look forward to the full publication.

Reviewer #2: (No Response)

7. PLOS authors have the option to publish the peer review history of their article (what does this mean?). If published, this will include your full peer review and any attached files.

Reviewer #1: No

Reviewer #2: No

---

## [Editor Report · Acceptance letter]

26 Oct 2021

PONE-D-21-01330R1 

Research sites get closer to field camps over time: Informing environmental management through a geospatial analysis of science in the McMurdo Dry Valleys, Antarctica 

Dear Dr. Chignell:

I'm pleased to inform you that your manuscript has been deemed suitable for publication in PLOS ONE. Congratulations! Your manuscript is now with our production department. 

Kind regards, 

on behalf of

Dr. Charles William Martin 

Academic Editor

PLOS ONE